# The impact of event type and geographical proximity on threat appraisal and emotional reactions to Wikipedia articles

Hannah Greving[1]*, Joachim Kimmerle[1,2]

**1** Knowledge Construction Lab, Leibniz-Institut für Wissensmedien, Tübingen, Germany, **2** Department of Psychology, Eberhard Karls University of Tübingen, Tübingen, Germany

* h.greving@iwm-tuebingen.de

**Data Availability Statement:** All relevant data are within the manuscript and its Supporting Information files.

**Funding:** The authors received no specific funding for this work.

## Abstract

The online encyclopedia Wikipedia strives for objectivity and neutrality. However, Wikipedia also provides articles about negative events (e.g., earthquakes, terrorist attacks) that likely elicit strong, negative emotions. These emotions might slip into Wikipedia articles. Previous research has demonstrated that Wikipedia articles on terrorist attacks contained more anger-related content than Wikipedia articles on earthquakes. This previous research focused on the expression of emotional reactions in existing Wikipedia articles and used an automatic linguistic analysis tool that counted the number of emotion-related words. In order to extend this approach, the first aim of the present research was to replicate these findings by focusing on the emotional reactions during and after reading the articles. Second, previous research did not look at the geographical location of the negative events, which may be a relevant, influential factor. Emotional reactions may be stronger for geographically closer events (i.e., Europe for Europeans) than for geographically more distant events (i.e., Asia). Two studies, one with few raters rating their emotional reactions to many Wikipedia articles (S1 Study) and another with many raters rating their emotional reactions to few Wikipedia articles (S1 Study), demonstrated that Wikipedia articles on terrorist attacks elicited more threat, anger, sadness, and anxiety than Wikipedia articles on earthquakes. These effects occurred for negative events in Europe but were absent for events in Asia, with one exception. The anger effect was the same across Europe and Asia. Thus, event type and geographical proximity are relevant factors for explaining threat and emotional reactions to Wikipedia articles.

## Introduction

The online encyclopedia Wikipedia is one of the most popular online services that people use to gain access to huge amounts of reliable information [1] (see also www.alexa.com). Besides factual information, Wikipedia is also a source of information about events that just happened, as anyone can contribute to Wikipedia anywhere and at any time. Devastating, negative events (e.g., earthquakes, terrorist attacks) belong to one category of these events. Although

**Competing interests:** The authors have declared that no competing interests exist.

Wikipedia strictly strives for unbiased, objective, and neutral information on its platform (https://en.wikipedia.org/wiki/Wikipedia:Neutral_point_of_view), such negative events have a high potential for eliciting strong emotions. Therefore, previous research has investigated whether existing Wikipedia articles differ in their emotional content [2–6]. This research found that Wikipedia articles on man-made attacks (e.g., terrorist attacks) contained more anger-related content than Wikipedia articles on nature-made disasters (e.g., earthquakes).

Yet, these previous studies leave venues for further research. The previous results were mostly based on the automatic linguistic text analysis tool Linguistic Inquiry and Word Count (LIWC [7]) and its mere counts of words. It remains unclear whether the effects found with LIWC are also present when people read the articles, and whether the content in the Wikipedia articles also elicits similar emotional reactions in its readers while and after reading the articles. Moreover, the previous research mostly analyzed English Wikipedia articles and did not account for the location where the negative event had happened. Yet, events that occur in close geographical proximity (e.g., in Europe for a European audience) may elicit stronger emotions than events that occur geographically further away (e.g., in Asia). Therefore, the current research investigated the emotional reactions to the content of existing Wikipedia articles—with either few raters rating their reactions to many Wikipedia articles or many raters rating their reactions to few Wikipedia articles—and controlled for the continent where the event had happened in a within-groups experiment. Thus, this research extends previous research in important ways and sheds new light on factors that are relevant for the elicitation of emotions by Wikipedia articles.

## Wikipedia and its characteristics

Wikipedia is the most frequently used online platform for acquiring information and, since its founding in 2001, has been growing ever since: It exists in more than 300 languages, of which the English Wikipedia is the most comprehensive one with more than 6 million articles (https://meta.wikimedia.org/wiki/List_of_Wikipedias). Thus, Wikipedia provides a huge information and knowledge database which is established by the effective and collective collaboration of masses of individuals on millions of articles [8]. In order to provide high quality information, Wikipedia has strict guidelines to guarantee the provision of unbiased, objective, and neutral information (https://en.wikipedia.org/wiki/Wikipedia:Neutral_point_of_view; [9–11]).

Despite these guidelines, Wikipedia articles may contain emotional content. Such emotional content is likely to be present in Wikipedia for three reasons. First, devastating, negative events such as earthquakes and terrorist attacks are the subject of numerous Wikipedia articles. Especially newcomers to the Wikipedia community often collaborate on articles about negative events that just happened and may not be altogether used to the guidelines of Wikipedia [10, 12–13]. That is why their emotions may subtly slip into the articles. Second, emotions are core to human beings and influence a wide array of aspects of human functioning and behavior [14]. In particular, they influence information processing [15–16], such as the production of information [17–18], which is relevant for writing Wikipedia articles. Across a variety of contexts, studies have shown that individuals produced more information that was congruent with their current state, for example, more negative information when they were in a negative state [18]. Third, and related to the previous point, it is known that characteristics of Internet users can spill over into Wikipedia and its content [19–21]. Research has shown, for example, that attitudes, opinions, prior knowledge, and group memberships are reflected in written Wiki texts [19, 22–24]. In that sense, emotions as subjective experiences and, thus, characteristics of Internet users are likely to spill over into Wikipedia's content in the same way.

## Emotional content in Wikipedia articles

Previous research has investigated whether Wikipedia articles about different negative events differ in their emotional content [3–5]. This research used appraisal theories of specific emotions [25–27] to explain the elicitation of certain emotions by certain negative events and used man-made attacks (e.g., terrorist attacks) and natural disasters (e.g., earthquakes) as negative events. These appraisal theories suggest that the characteristics of a situation are important for which emotions are elicited. Emotions can be elicited by desirable or undesirable outcomes, actions, or objects. As we investigate negative events, we will focus on undesirable outcomes and actions. Undesirable outcomes that cannot be undone and have unspecific causes elicit sadness [25–27]. In contrast, undesirable outcomes that are caused by harmful and blameworthy actions of others who intended to do harm and can be held responsible elicit anger. The previous research mapped these predictions of appraisal theories onto the natural disasters and man-made attacks. In particular, terrorist attacks were expected to elicit more anger, as they occur due to the blameworthy actions of extremists, while earthquakes were expected to elicit more sadness, as they occur due to natural factors and cannot be undone [25–27] (for more elaboration on these expectations see [5]). Following this theorizing and these expectations, the previous research analyzed existing Wikipedia articles in order to investigate whether those expected emotional reactions were present in the manifest content of the articles. It was found that Wikipedia articles on man-made attacks contained more anger-related content than Wikipedia articles on natural disasters, whereas Wikipedia articles on natural disasters contained more sadness-related content than Wikipedia articles on man-made attacks [4–5].

Besides the research on the content of existing Wikipedia articles, there has also been research that investigated whether these effects hold when individuals spontaneously produce a Wikipedia article about these negative events. That research found that participants also produced more anger-related content in Wikipedia articles on man-made attacks than in articles on natural disasters in controlled laboratory experiments [3]. These laboratory experiments also assessed participants' emotional reactions with self-report measures and further examined what could explain the robust anger effects, both as content in the Wikipedia articles and as self-reported reactions. Based on theories of threat [28], it was argued that man-made attacks elicit threat appraisal and as a consequence anger. Threat appraisal is defined as feelings of being unable to deal with a demanding situation [29]. Threat theories suggest that threat elicits active and engaged reactions, such as increased behavioral intentions and self-serving reactions [28]. It has also been shown that terrorist attacks were perceived as threatening and increased support for engagement in war activities and aggressive reactions [30–32]. Based on these theories and findings, the previous research has demonstrated that man-made attacks elicited more threat appraisal than nature-made disasters. Threat appraisal was also a mediator that could explain the effects on anger and anger-related content in Wikipedia articles.

This previous research investigated which manifest content existing Wikipedia articles about negative events have and which information is produced when spontaneously writing those Wikipedia articles. Thus, the research focused on the expression of emotions in the articles. What has been neglected so far is how information from the Wikipedia articles is received and which emotional reactions are elicited when reading them. These steps are quite important elements in the communication process of Wikipedia, as Wikipedia articles are produced and written for other Internet users who receive its content and react to it emotionally.

Second, the Wikipedia articles or texts in the previous studies were analyzed with the automatic linguistic text analysis software Linguistic Inquiry and Word Count (LIWC [7]). This software is the most frequently used text analysis tool [33–36]. It is based on the dictionary of a

certain language and counts the number of words which belong to certain categories that had been defined by linguistic experts beforehand. One category includes the negative emotion categories anger, sadness, and anxiety, and these were used in the previous research. Although widely used, LIWC can be criticized for being based merely on the counts of words. LIWC is not able to assess any context that these emotional words are presented in, any relationships between emotional words, which may hint to totally different emotions (e.g., in the case of sarcasm or irony), or any political or societal meaning of acts or words (e.g., in the case of terrorist attacks). Thus, the results obtained in previous research for the emotional content in the Wikipedia articles may not completely hold, because LIWC may not entirely capture anger, sadness, and anxiety and may not consider the context or other relevant factors. It is also unclear whether individuals reading the Wikipedia articles react emotionally in the same way as indicated by LIWC. In order to close this gap, the current research set out to investigate emotional reactions to the content of Wikipedia articles on negative events.

Another question that could be drawn from previous research is whether the geographical location (e.g., the continent) of the negative event should be considered. In this previous research, only English Wikipedia articles were analyzed. This means that the articles could have been written anywhere by anyone who speaks English. Consequently, English language articles do not allow for taking into account how close or distant the negative event had been for the Internet users who contributed to these articles. From other research, it is known that the geographical proximity of a negative event is quite relevant for the reaction to the event and for its coverage in the media. Close events elicited strong emotions [37] and there was more intense reporting about close events than distant events [38–40]. Such intense reporting could be an amplifier of the strong emotions. In this sense, investigating the geographical proximity of an event is a relevant factor for the elicitation of emotions, with closer events eliciting stronger emotions than more distant events.

## The current research

The aim of this research was to shed new light on the elicitation of emotions and threat appraisal by Wikipedia articles. In particular, in this research raters rated their emotional reactions during and after reading the content of Wikipedia articles. We conducted two studies, each with a different focus. In Study 1, four raters rated 60 existing Wikipedia articles on earthquakes and terrorist attacks regarding their emotional reactions while reading the articles. As a conceptual extension, in Study 2, 35 participants serving as independent raters indicated their emotional reactions after reading four existing Wikipedia articles on earthquakes and terrorist attacks. Moreover, Study 2 used an Asian and a European earthquake as well as an Asian and a European terrorist attack in order to take the geographical proximity of the negative event into account. Thus, the studies of the current research complement each other by focusing either on many articles with a few raters or a few articles with many raters, and by shifting the rating focus from rating emotional reactions while reading the text to capturing emotional reactions after reading.

Based on the theoretical considerations outlined above and on the findings of previous research [3–5], we stated the following hypotheses regarding main effects of type of negative event:

Wikipedia articles on terrorist attacks elicit more threat appraisal than Wikipedia articles on earthquakes (*Hypothesis 1*). Wikipedia articles on terrorist attacks elicit more anger than Wikipedia articles on earthquakes (*Hypothesis 2*). Wikipedia articles on earthquakes elicit more sadness than Wikipedia articles on terrorist attacks (*Hypothesis 3*). As an open research question, we also examined the impact of the type of negative event on anxiety (for more elaboration on this point see [5]).

Moreover, we stated the following hypotheses regarding interaction effects between type of negative event and geographical proximity:

When they occurred geographically more closely, terrorist attacks and earthquakes differ more strongly in their elicitation of threat appraisal (*Hypothesis 4a*), anger (*Hypothesis 4b*), and sadness (*Hypothesis 4c*) than when they occurred geographically further away. As an open research question, we also considered whether there are respective effects for anxiety (for more elaboration on this point see [5]).

Study 1 investigated 30 existing Wikipedia articles on earthquakes and 30 existing Wikipedia articles on terrorist attacks and tested Hypotheses 1–3. The elicitation of emotions while reading the articles was measured by having four independent raters rate their subjective emotional reactions during reading.

## Study 1

### Method

**Design, power analysis, materials, and procedure.**    The studies reported here were reviewed by the local ethics committe of the Leibniz-Institut für Wissensmedien. The approval number is LEK 2016/046. We received written consent from participants and used a standard consent form to inform participants about the study. Yet, participants' data was analyzed anonymously. Study 1 had a one-factorial between-groups design with two conditions (type of negative event: earthquakes vs. terrorist attacks). We conducted an a priori power analysis with G∗Power [41] for a one-way analysis of variance (ANOVA) with two groups with a medium to large effect size of $\rho = 0.40$, a power of 0.80, and an alpha level of $\alpha = 0.05$. This analysis showed that we would need a total sample size of 52 Wikipedia articles. Following these calculations and ensuring to have enough power, we used 60 Wikipedia articles in total, that is, 30 Wikipedia articles about earthquakes and 30 Wikipedia articles about terrorist attacks (see Appendix; see also [5]). The selection criteria for these articles were that they dealt with the respective event, were available in English, and did not occur earlier than 2001, because Wikipedia has only existed since then. Another criterion relevant for emotional reactions was that the article was initially created immediately after the event had happened. From these 30 articles about earthquakes and 30 articles about terrorist attacks, we only used the main text and excluded the tables, figures, notes, references, and external links in order to prevent confounds stemming from material different from the main article.

In order to rate the emotional reactions to the articles, we used four independent raters. Their task was to read each article and rate their own emotional reactions while reading. In particular, they were instructed to spontaneously indicate their subjective emotional reactions after each large paragraph within the Wikipedia articles. On average, the articles had $M = 5.63$ paragraphs ($SD = 2.35$, range: 1–11). For each rater, the ratings per paragraph were later averaged across each article. Regarding the emotional reactions, the four raters indicated how threatened ($\alpha = .27$), angry ($\alpha = .76$), sad ($\alpha = .55$), and anxious ($\alpha = .42$) they felt on a 9-point Likert scale ranging from 1 (*not at all*) to 9 (*very much*).

### Results

We tested whether Wikipedia articles about earthquakes and terrorist attacks differed in the emotional reactions they elicited while the raters were reading them. To do so, we conducted separate one-way ANOVAs with negative event (earthquakes vs. terrorist attacks) as between-groups factor for threat, anger, sadness, and anxiety. The results revealed that there was an effect for *threat*, $F(1, 58) = 12.75$, $p = .001$, $\eta_p^2 = .180$. Wikipedia articles about terrorist attacks elicited more feelings of threat ($M = 4.39$, $SD = 0.70$) than Wikipedia articles about

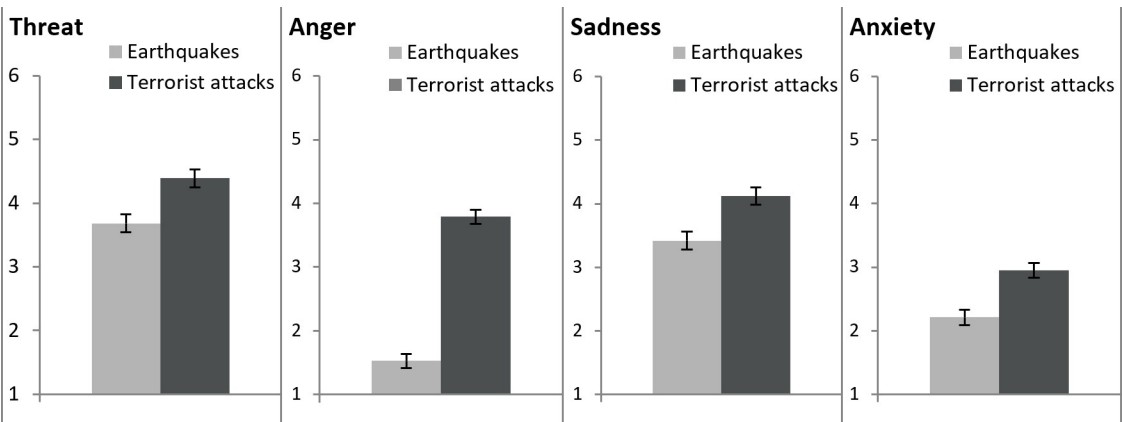

**Fig 1. Means and standard errors for the dependent variables threat, anger, sadness, and anxiety for the Wikipedia articles on earthquakes and the Wikipedia articles on terrorist attacks of Study 1.**

earthquakes ($M = 3.68$, $SD = 0.84$; Fig 1 on the far left), which supported Hypothesis 1 and was consistent with previous research [3]. The effect for *anger* was significant as well, $F(1, 58) = 197.45$, $p < .001$, $\eta_p^2 = .773$. Wikipedia articles about terrorist attacks elicited more anger ($M = 3.79$, $SD = 0.69$) than Wikipedia articles about earthquakes ($M = 1.53$, $SD = 0.55$; Fig 1 second from the left), which supported Hypothesis 2 and was in line with previous findings [3–5]. There was also a significant effect for *sadness*, $F(1, 58) = 11.64$, $p = .001$, $\eta_p^2 = .167$. Wikipedia articles about terrorist attacks elicited more sadness ($M = 4.12$, $SD = 0.70$) than Wikipedia articles about earthquakes ($M = 3.42$, $SD = 0.87$; Fig 1 second from the right), which was opposite to Hypothesis 3 and found in some research [3] but not in others [4–5]. Finally, the effect for *anxiety* was significant as well, $F(1, 58) = 18.98$, $p < .001$, $\eta_p^2 = .247$. Wikipedia articles about terrorist attacks elicited more anxiety ($M = 2.95$, $SD = 0.69$) than Wikipedia articles about earthquakes ($M = 2.21$, $SD = 0.61$; Fig 1 on the far right). This result could also be found in some [3] but not in other research [4–5].

## Discussion

The findings of Study 1 demonstrated that reading Wikipedia articles on terrorist attacks elicited more feelings of threat, anger, sadness, and anxiety than reading Wikipedia articles on earthquakes. The findings for threat and anger are in line with previous research [3–5]. The ratings of the independent raters were at the lower end of the 9-point scale. Probably, merely reading about the negative events did not elicit intensive levels of threat, anger, sadness, and anxiety but rather low to medium levels. This may be the case because most of the events had happened in the past and may not have as much influence as new events. Still, the findings clearly demonstrated the expected differences between the negative events. The level and pattern of results also mainly corresponded to previous research [3–5]. Nonetheless, one can easily imagine that the level of emotions would very likely increase in the face of a new terrorist attack or earthquake (e.g., when hearing about a new negative event in the news or when even witnessing or experiencing such an event). Still, the expected difference between terrorist attacks and earthquakes should hold in this case as well.

One could argue that although many Wikipedia articles were rated, the results of Study 1 were based on a relatively small number of raters and partly suboptimal reliabilities. In order to improve our method, we used more elaborate measurements for threat and the three emotions and changed the following aspects in Study 2: (1) We used only two Wikipedia articles

on earthquakes and two Wikipedia articles on terrorist attacks, (2) we used more raters, that is, 35 participants who rated these articles and used a within-groups design, (3) we instructed participants to indicate their emotional reactions after reading the whole Wikipedia article, and (4) we applied a commonly used appraisal measure for threat (i.e., a difference score between experienced demands and resources) and assessed the emotions with three items each.

Moreover, we added geographical proximity (i.e., the location where the negative event had happened) as a factor to our study design. The study was conducted in Germany, which means that negative events happening in Europe were geographically closer than negative events happening in Asia. Consequently, the type of a negative event (i.e., an earthquake or a terrorist attack) should have a greater impact when the events occurred in Europe compared to when they occurred in Asia. Thus, in Study 2, we tested Hypotheses 1–3 and Hypotheses 4a–4c.

Study 2 used four existing Wikipedia articles about an earthquake in Asia, an earthquake in Europe, a terrorist attack in Asia, and a terrorist attack in Europe. We investigated the emotional reactions of participants after they had read each one of the four articles and used a complete within-groups design.

## Study 2

### Method

**Design, power analysis, and participants.**   The study had a 2 (type of negative event: earthquake vs. terrorist attack) × 2 (geographical proximity: Asia vs. Europe) within-groups design. We conducted an a priori power analysis with G*Power [41] for a repeated measures ANOVA with four measurement points with a medium to large effect size of $\rho = 0.40$, a power of 0.80, and an alpha level of $\alpha = 0.05$. This analysis showed that we would need a total sample size of 10 participants. We followed these calculations, but simultaneously had to adjust to the conditions in the laboratory and the availability of participants. As participants had signed up for a whole lab session (see below), we could not simply stop the study after the 10 necessary participants. Therefore, more participants than needed participated in this study, that is, 35 participants (26 female, 9 male; $M_{age} = 26.91$, $SD = 9.94$) who had German as mother tongue took part in the study. The study lasted approximately 20 minutes and was conducted as the second study of a one-hour lab session. For their participation in the entire one-hour lab session, participants received €8 (approximately $9) as compensation.

**Procedure and materials.**   The study was conducted in a laboratory. This laboratory comprised six private cubicles which were soundproof and each equipped with a computer. The study was conducted with the computers and, thus, all instructions and measures were displayed on the computer screens. After giving informed consent, all participants read four existing German Wikipedia articles on negative events. We provided German articles as we wanted to keep the language of the laboratory study entirely in German. Moreover, the German articles may have been more easily comprehensible for participants. These articles only contained the main text and did not include figures, tables, references, external links, or other information unrelated to the main text. The criteria for the selection of the articles were the same as in Study 1, but we applied the following additional criteria: The articles were available in German, the lengths of the main texts were comparable across geographical location, and the event had happened in the last few years. These additional criteria yielded a sample of roughly 10 articles. From these articles, we then chose those four articles whose length of the German texts was sufficiently long to ensure that participants could engage with the negative events long enough. Finally, we used the following articles: April 2015 Nepal earthquake (earthquake in Asia), August 2016 earthquake in Italy (earthquake in Europe), January 2016 Bacha Khan University attack (terrorist attack in Asia), and June 2017 London Bridge attack (terrorist attack in

Europe). Two of these articles (April 2015 Nepal earthquake, January 2016 Bacha University attack) have also been used in their English version in Study 1.

Roughly half of the participants ($N = 18$) first read the two earthquakes articles and then the two terrorist attacks articles, while the other half of the participants ($N = 17$) first read the two terrorist attacks articles and then the two earthquakes articles. We checked whether this difference in the order of articles had an influence on the dependent variables. To do so, we conducted four mixed ANOVAs with order of articles as between-groups factor and the four ratings of the articles of each dependent variable (i.e., threat appraisal, anger, sadness, and anxiety) as within-groups factor. These analyses did not reveal any significant main effect of order of articles on the dependent variables, all $F$'s (1, 33) < 1, *ns*. Therefore, we excluded the order factor from the analyses.

After reading each Wikipedia article, participants, first, indicated how threatening they appraised the event described in the Wikipedia article and, then, which emotions they felt concerning this event. To make sure that participants would leave the laboratory in a balanced state of mood, they thought about a current positive situation in their lives at the end of the study and wrote some sentences about it. Finally, they were carefully debriefed, received their compensation, and were dismissed.

**Measures.** *Threat appraisal* was measured with two items, a demands item and a resources item. The demands item asked participants "How demanding is the negative event in the article you just read for you?" and assessed their answers on a 9-point Likert scale ranging from 1 (*not demanding at all*) to 9 (*very demanding*). The resources item asked participants "How well can you deal with the negative event?" and assessed their answers on a 9-point Likert scale ranging from 1 (*not at all*) to 9 (*very well*). These items are typical items often used in research about threat [29, 42]. We then calculated a difference score of the two items by subtracting the scores of the resources item from the scores of the demands item, which resulted in a single score for threat appraisal. Thus, this difference score could take positive as well as negative values with higher values indicating more threat appraisal.

As *emotions*, we measured with three items each *anger* (hostile, angry, annoyed, α = .77), *sadness* (disappointed, sad, depressed, α = .94), and *anxiety* (scared, afraid, frightened, α = .94; [43]) regarding the negative event in the article. All items were assessed on a 9-point Likert scale ranging from 1 (*does not apply at all*) to 9 (*applies very much*) and were preceded by the words "When I am thinking about the negative event in the article, I am/feel . . .".

## Results

In order to test whether the four articles differed in their elicitation of threat appraisal, anger, sadness, and anxiety, we conducted repeated measurement ANOVAs with type of negative event (earthquake vs. terrorist attack) and geographical proximity (Asia vs. Europe) as within-groups factors. We conducted these ANOVAs for each of the dependent variables in the same way and, in the following, present the results of these analyses for each variable.

**Threat appraisal.** The results revealed a significant main effect of type of negative event, $F$ (1, 34) = 5.64, $p = .023$, $\eta_p^2 = .142$. The terrorist attacks elicited more threat appraisal ($M = 0.50$, $SD = 2.92$) than the earthquakes ($M = -0.39$, $SD = 2.76$), which supported Hypothesis 1. The results also revealed a significant type of negative event × geographical proximity interaction, $F$ (1, 34) = 21.72, $p < .001$, $\eta_p^2 = .390$. The threat appraisal of the earthquake in Asia ($M = 0.11$, $SD = 2.83$) did not differ from the threat appraisal of the terrorist attack in Asia ($M = -0.40$, $SD = 3.14$; $M_{\text{diff}} = 0.51$, $SE = 0.52$, $p = .332$). In contrast, the terrorist attack in Europe elicited more threat appraisal ($M = 1.40$, $SD = 2.66$) than the earthquake in Europe ($M = -0.89$, $SD = 2.69$; $M_{\text{diff}} = 2.29$, $SE = 0.43$, $p < .001$; Fig 2 on the far left), which supported Hypothesis 4a. The main effect of geographical proximity was not significant, $F$ (1, 34) = 1.53, $p = .225$.

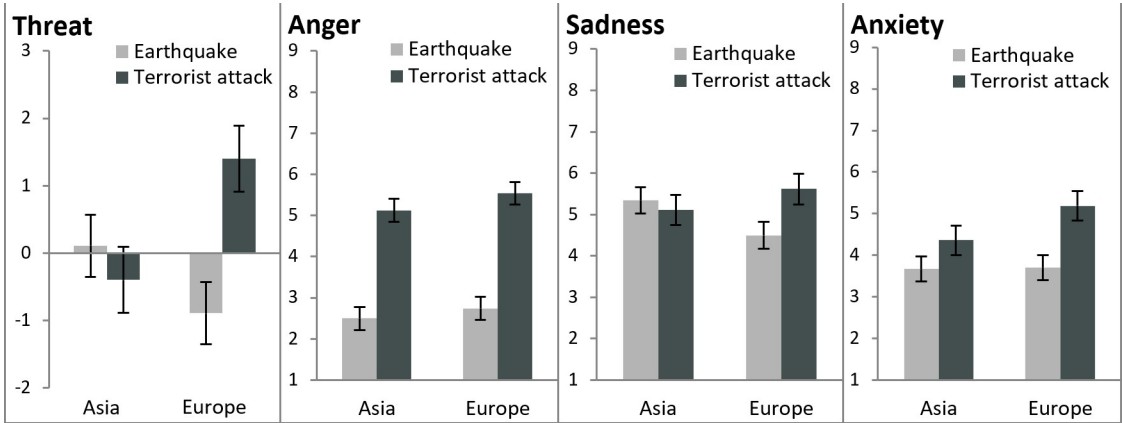

**Fig 2. Means and standard errors for the dependent variables threat, anger, sadness, and anxiety for the Wikipedia articles on earthquakes and terrorist attacks that happened in Asia or Europe of Study 2.**

**Anger.** The results revealed a significant main effect of type of negative event, $F(1, 34) = 117.46, p < .001, \eta_p^2 = .776$. The terrorist attacks elicited more anger ($M = 5.33, SD = 1.68$) than the earthquakes ($M = 2.62, SD = 1.68$; Fig 2 second from the left), which supported Hypothesis 2. There was also a marginal main effect of geographical proximity, $F(1, 34) = 4.04, p = .052, \eta_p^2 = .106$. The negative events in Europe elicited more anger ($M = 4.14, SD = 1.76$) than the negative events in Asia ($M = 3.81, SD = 1.60$). The interaction was not significant, $F(1, 34) < 1, ns$. Thus, Hypothesis 4b was not supported. Instead, Hypothesis 2 was supported for terrorist attacks in both Asia and Europe.

**Sadness.** The results revealed a significant main effect of type of negative event, $F(1, 34) = 5.29, p = .028, \eta_p^2 = .135$. The terrorist attacks elicited more sadness ($M = 5.37, SD = 2.19$) than the earthquakes ($M = 4.92, SD = 1.93$), which was opposite to Hypothesis 3. There was also a significant type of negative event × geographical proximity interaction, $F(1, 34) = 25.45, p < .001, \eta_p^2 = .428$. The earthquake in Asia ($M = 5.34, SD = 2.04$) did not differ from the terrorist attack in Asia ($M = 5.11, SD = 2.33$) in the elicitation of sadness ($M_{diff} = 0.23, SE = 0.26, p = .383$). In contrast, the terrorist attack in Europe elicited more sadness ($M = 5.62, SD = 2.05$) than the earthquake in Europe ($M = 4.51, SD = 1.81; M_{diff} = 1.11, SE = 0.21, p < .001$; Fig 2 second from the right), which supported Hypothesis 4c, but was still opposite to Hypothesis 3. The main effect for geographical proximity was not significant, $F(1, 34) < 1, ns$.

**Anxiety.** The results revealed that there were significant main effects for type of negative event, $F(1, 34) = 27.27, p < .001, \eta_p^2 = .445$, and geographical proximity, $F(1, 34) = 9.23, p = .005, \eta_p^2 = .213$, which were qualified by a significant type of negative event × geographical proximity interaction, $F(1, 34) = 8.02, p = .008, \eta_p^2 = .191$. The terrorist attack in Asia elicited more anxiety ($M = 4.36, SD = 1.91$) than the earthquake in Asia ($M = 3.67, SD = 1.76; M_{diff} = 0.69, SE = 0.29, p = .023$). The terrorist attack in Europe elicited even more anxiety ($M = 5.19, SD = 2.20$) than the terrorist attack in Asia ($M_{diff} = 0.83, SE = 0.17, p < .001$) and also more anxiety than the earthquake in Europe ($M = 3.70, SD = 1.86; M_{diff} = 1.49, SE = 0.21, p < .001$; Fig 2 on the far right).

## Discussion

The findings of Study 2 showed that the Wikipedia article on the nearby (i.e., European) terrorist attack elicited more threat appraisal than the Wikipedia article on the nearby earthquake, which was not the case when the negative events happened far away (i.e., in Asia). For

the elicitation of anger, however, the geographical proximity of the negative event did not matter. The Wikipedia articles on the nearby and the far away terrorist attacks both elicited more anger than the Wikipedia articles on the nearby and the far away earthquakes. Concerning sadness and anxiety, the Wikipedia article on the nearby terrorist attack elicited more sadness and anxiety than the Wikipedia article on the nearby earthquake. There was no difference in sadness for the Wikipedia articles on the far away earthquake or terrorist attacks, but there was a difference in anxiety.

Similar to Study 1, the scores of the emotions anger, sadness, and anxiety were around the midpoint or lower end of the 9-point scale. As explained in the discussion of Study 1, these moderate scores may have occurred as participants had merely read about the negative events. Possibly, hearing about or watching news on television about new negative events, or even witnessing or experiencing these events may intensely increase these emotions. Still, the difference between earthquakes and terrorist attacks should also hold in such situations.

## General discussion

The findings of the current research present evidence that Wikipedia articles on terrorist attacks elicit more threat appraisal and anger than Wikipedia articles on earthquakes. In particular, these emotions were elicited while reading as well as after reading these articles. Moreover, Wikipedia articles on terrorist attacks also elicited more sadness and anxiety while reading and after reading them than Wikipedia articles on earthquakes.

This research complements existing research in three important ways. First of all, previous research on emotional content in Wikipedia articles [3–5] used the automatic linguistic text analysis tool LIWC to investigate emotional content in Wikipedia articles. Although widely used [33–36], this tool may fall short in considering the context, the relationships between emotional words, or the meaning of acts or events. The present research adds to such previous research by having individuals rate their emotional reactions while and after reading the Wikipedia articles. Similar to previous research, the studies presented here found effects for threat appraisal and anger.

Second, the present research closes an important research gap. Previous research has demonstrated that Wikipedia articles written collectively by masses of individuals as well as individually written Wikipedia articles in the laboratory contained more anger-related content in the case of man-made attacks (e.g., terrorist attacks) and mostly more sadness-related content in the case of disasters (e.g., earthquakes, train accidents). The previous laboratory research also measured emotional reactions and threat separately with self-report measures and demonstrated that merely thinking about a hypothetical man-made attack elicited more anger and a higher threat appraisal than thinking about a hypothetical man-made or nature-made disaster. The present studies bring these two lines of research together by illustrating that such emotions were also elicited both while reading and after reading Wikipedia articles on real negative events. Importantly, the same effects as in previous research were found. As such, previous research and the current research demonstrated together that the threat appraisal and anger effects occurred both while externalizing information (i.e., contributing information to Wikipedia articles) [3–5] and internalizing information (i.e., reading and encoding information from Wikipedia articles).

Finally, the present research introduced a new factor that previous research did not consider. That is, it investigated the geographical proximity of the negative events. In line with other research on the media coverage of certain events [38–39], we found differences between the terrorist attack and the earthquake for threat appraisal, sadness, and anxiety when the events had happened close by. This was not the case when the events happened further away.

This means that for threat appraisal, sadness, and anxiety, it mattered where the negative event had happened. The closer-by it happened, that is, in Europe, the stronger were the emotional reactions and threat responses. In contrast, the distance to Asia seemed to neutralize and attenuate the reactions. For anger, however, it did not matter whether the terrorist attacks were close by or further away. They both elicited more anger than close-by or further away earthquakes. In this sense, the distance of the negative event had no neutralizing or attenuating effect on the anger effects. Rather, the anger effects for man-made attacks seemed to be independent from geographical proximity. Possibly, the fact that extremists increasingly aim to do harm across the whole world makes terrorist attacks a more global phenomenon than earthquakes that are tied to specific regions. These differences may have caused the particular anger effects.

## Implications

In the same way as in previous research, the findings of this research imply that appraisal theories of specific emotions [25–27], theories of threat appraisal [28–29], and influences of emotions on information processing [17–18] can be applied to Wikipedia. Specific characteristics of certain situations, such as the characteristics of a negative event, elicit specific emotions. That is, reading about the blameworthy actions of terrorists in Wikipedia articles elicits anger and threat appraisal. Thus, those theories are relevant for information production in and knowledge acquisition from Wikipedia articles.

The results of the present research extend previous research in an important way, as they also investigated geographical proximity as a factor influencing emotional reactions to Wikipedia articles. We revealed that geographical proximity has an impact on threat appraisal, sadness and anxiety, while it has no influence on anger. Thus, regarding theories of specific emotions [25–27], not only the situation of the negative event but also the geographical proximity may have to be taken into account to specify the emotional reaction. Moreover, for Wikipedia researchers and the Wikipedia community, it is relevant to know that negative events happening further away may elicit weaker emotional reactions than events happening closer by. This finding may have practical implications for Wikipedia authors as well as for researchers studying emotions in Wikipedia.

## Strengths, limitations, and future research

The present research used existing Wikipedia articles. Raters rated their emotional reactions to those articles in two studies that complement each other. While Study 1 used various Wikipedia articles and four raters, Study 2 used four articles but had numerous raters. In this sense, the studies were well set up and worked well together. They also introduced a new factor, that is, geographical proximity, which previous research had not taken into consideration so far.

Still, some limitations can be discussed. First, the articles in Study 1 were in English to ensure comparability with previous research [4–5], while the articles of Study 2 were in German in order to keep the material of the entire laboratory study in German. Such language differences may have caused different results. The raters of the two studies were also different. While they were research assistants collaborating with the authors in Study 1, they were usual laboratory participants in Study 2. Yet, all raters of Study 1 and most raters of Study 2 were university students and therefore comparable. Moreover, the Wikipedia articles were different across the studies. There were more articles in Study 1 than in Study 2. Still, from the four articles of Study 2, one earthquake and one terrorist attack were also used in Study 1. In that sense, there was some overlap in terms of Wikipedia articles between the studies. Besides these differences, the two studies worked well together, but future research should nonetheless strive to minimize possible differences.

Second, in Study 2, the factor geographical proximity was established by having four Wikipedia articles, two about negative events in Asia and two about negative events in Europe. Although the effects in this study were clear, other research should aim to replicate these findings with more events. Such research could also specify geographical proximity in a more fine-grained way. For example, it could be differentiated between very close and close negative events in Europe, negative events in Asia which happened just outside of Europe, and further away negative events in Asia. This would yield a more detailed picture of the influence of geographical proximity.

Third, the studies of this research were rating studies which used a typical rating approach or a complete within-groups design in the laboratory. Yet, in future research, the current findings should be replicated in controlled laboratory studies with between-groups factors. Such studies would enable researchers to ensure that the present findings also hold for more conservative statistical tests in between-groups designs.

Finally, the current research found effects for emotional reactions while reading and after reading Wikipedia articles. In order to investigate whether individuals' physiological reactions are the same as the reactions our studies demonstrate, future research could use physiological measures. For example, participants could wear EEG caps that assess their brain activity while reading a Wikipedia article about a terrorist attack. Likewise, physiological measures such as heart rate, blood pressure, skin conductance, and pupil dilatation could be highly relevant additional measures in assessing emotional reactions. If such future research found similar results as we did, it could strengthen the current findings and, therefore, provide further relevant physiological insights about emotional reactions to Wikipedia articles about negative events.

## Conclusion

In sum, the current research has demonstrated that Wikipedia articles on terrorist attacks elicit more anger and threat appraisal than Wikipedia articles on earthquakes. These effects were established with two rating studies that varied the number of Wikipedia articles and the number of raters. This research also illustrated that the emotion effects only occur for negative events in Europe and mostly not for negative events in Asia, except for the anger effect which was the same across Europe and Asia. Thus, the type of negative event and the geographical proximity of the negative event are relevant factors for explaining threat appraisal responses and emotional reactions to Wikipedia articles.

## Supporting information

**S1 Appendix.**
(DOCX)

**S1 Study.**
(SAV)

**S2 Study.**
(SAV)

## Acknowledgments

We wish to kindly thank Maxi Bürkle, Hannah Engelhardt, Laura Hertner, and Lisa Rau for their help with content rating, study preparation, and data collection

## Author Contributions

**Conceptualization:** Hannah Greving.

**Data curation:** Hannah Greving.

**Formal analysis:** Hannah Greving.

**Investigation:** Hannah Greving.

**Methodology:** Hannah Greving, Joachim Kimmerle.

**Supervision:** Joachim Kimmerle.

**Validation:** Hannah Greving, Joachim Kimmerle.

**Visualization:** Hannah Greving.

**Writing – original draft:** Hannah Greving.

**Writing – review & editing:** Hannah Greving, Joachim Kimmerle.

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
