## [Editor Report · Decision Letter 0]

17 Oct 2019

PONE-D-19-26386

The impact of event type and geographical proximity on threat appraisal and emotions in Wikipedia articles

PLOS ONE

Dear Dr. Greving,

Thank you for submitting your manuscript to PLOS ONE. After careful consideration, we feel that it has merit but does not fully meet PLOS ONE’s publication criteria as it currently stands. Therefore, we invite you to submit a revised version of the manuscript that addresses the points raised during the review process.

We would appreciate receiving your revised manuscript by Dec 01 2019 11:59PM. To enhance the reproducibility of your results, we recommend that if applicable you deposit your laboratory protocols in protocols.io, where a protocol can be assigned its own identifier (DOI) such that it can be cited independently in the future. For instructions see: http://journals.plos.org/plosone/s/submission-guidelines#loc-laboratory-protocols

We look forward to receiving your revised manuscript.

Kind regards,

Dana Balas Timar

Academic Editor

PLOS ONE

Journal Requirements:

Additional Editor Comments (if provided):

The authors present interesting data shedding new light on factors that are relevant for the elicitation of emotions by Wikipedia articles, namely Wikipedia articles on terrorist attacks elicited more threat, anger, sadness, and anxiety than Wikipedia articles on earthquakes.

The authors present to studies. The findings of Study 1 demonstrated that reading Wikipedia articles on terrorist attacks elicited more feelings of threat, anger, sadness, and anxiety than reading Wikipedia articles on earthquakes. The findings of Study 2 showed that the Wikipedia article on the nearby (i.e., European) terrorist attack elicited more threat appraisal than the Wikipedia article on the nearby earthquake, which was not the case when the negative events happened far away (i.e., in Asia).

The manuscript is well written, has an important threat appraisal message, and should be of great interest to the readers.

However, the results from Study 1 are somehow restrictive in amplitude, due to several aspects:

• “Wikipedia articles about terrorist attacks elicited more feelings of threat (M = 4.39, SD = 0.70) than Wikipedia articles about earthquakes (M = 3.68, SD = 0.84; Fig 1 on the far left), which supported Hypothesis 1”

• “Wikipedia articles about terrorist attacks elicited more anger (M = 3.79, SD = 0.69) than Wikipedia articles about earthquakes (M = 1.53, SD = 0.55; Fig 1 second from the left), which supported Hypothesis 2.”

• Wikipedia articles about terrorist attacks elicited more sadness (M = 4.12, SD = 0.70) than Wikipedia articles about earthquakes (M = 3.42, SD = 0.87; Fig 1 second from the right), which was opposite to Hypothesis 3.

• Wikipedia articles about terrorist attacks elicited more anxiety (M = 2.95, SD = 201 0.69) than Wikipedia articles about earthquakes (M = 2.21, SD = 0.61; Fig 1 on the far right).

when using a 9-point Likert scale ranging from 1 (not at all) to 9 (very much). The means are closer to the non-threat interval (1) rather that being closer to the threat interval (9).

We ask authors to explain such results and to further investigate the meaning of these low means on aspects considered to be important in the investigation of eliciting emotions.

Similar for Study 2, plus the data presenting threat appraisal from lines 292-301 it is inconsistent with results from Study 1. For example “The terrorist attacks elicited more threat appraisal (M = 0.50, SD = 2.92) than the earthquakes (M = -0.39, SD = 2.76), which supported Hypothesis 1.” Means and standard deviations are not properly presented, a mean is not supposed to be negative under the presented methodology. Please revise this section.

Overall, it is an important study, and should be considered for publication in PLOS ONE, once the statistical issue has been resolved.
---

## [Author Response · Author response to Decision Letter 0]

24 Oct 2019

Dear Dr. Balas Timar,

Thank you very much for investing your time and effort to review our manuscript. Your valuable feedback helped us improve the paper on its way to publication.

We have considered all your suggestions and are convinced that we have made appropriate improvements. In this letter, we address all your comments and explain our revisions in detail. In order to ensure the transparency of our revisions, we have replied to each comment separately (in bold letter type). Below our comments, you find the new or modified paragraphs of the revised manuscript (indented, in quotation marks, and in bold letter type).

Additional Editor Comments (if provided):

The authors present interesting data shedding new light on factors that are relevant for the elicitation of emotions by Wikipedia articles, namely Wikipedia articles on terrorist attacks elicited more threat, anger, sadness, and anxiety than Wikipedia articles on earthquakes.

The authors present to studies. The findings of Study 1 demonstrated that reading Wikipedia articles on terrorist attacks elicited more feelings of threat, anger, sadness, and anxiety than reading Wikipedia articles on earthquakes. The findings of Study 2 showed that the Wikipedia article on the nearby (i.e., European) terrorist attack elicited more threat appraisal than the Wikipedia article on the nearby earthquake, which was not the case when the negative events happened far away (i.e., in Asia).

The manuscript is well written, has an important threat appraisal message, and should be of great interest to the readers.

Thank you very much. We appreciate your positive feedback to our manuscript.

However, the results from Study 1 are somehow restrictive in amplitude, due to several aspects:

• “Wikipedia articles about terrorist attacks elicited more feelings of threat (M = 4.39, SD = 0.70) than Wikipedia articles about earthquakes (M = 3.68, SD = 0.84; Fig 1 on the far left), which supported Hypothesis 1”

• “Wikipedia articles about terrorist attacks elicited more anger (M = 3.79, SD = 0.69) than Wikipedia articles about earthquakes (M = 1.53, SD = 0.55; Fig 1 second from the left), which supported Hypothesis 2.”

• Wikipedia articles about terrorist attacks elicited more sadness (M = 4.12, SD = 0.70) than Wikipedia articles about earthquakes (M = 3.42, SD = 0.87; Fig 1 second from the right), which was opposite to Hypothesis 3.

• Wikipedia articles about terrorist attacks elicited more anxiety (M = 2.95, SD = 201 0.69) than Wikipedia articles about earthquakes (M = 2.21, SD = 0.61; Fig 1 on the far right).

when using a 9-point Likert scale ranging from 1 (not at all) to 9 (very much). The means are closer to the non-threat interval (1) rather that being closer to the threat interval (9).

We ask authors to explain such results and to further investigate the meaning of these low means on aspects considered to be important in the investigation of eliciting emotions.

This is truly a valid point. We have extended the discussion section of Study 1; we state that the means are at the lower end of the scale and we discuss why this could be the case.

“The ratings of the independent coders were at the lower end of the 9-point scale. Probably, merely reading about the negative events did not elicit intensive levels of threat, anger, sadness, and anxiety but rather low to medium levels. This may be the case because most of the events had happened in the past and may not have as much influence as new events. Still, the findings clearly demonstrated the expected differences between the negative events. The level and pattern of results also mainly corresponded to previous research [2-4]. Nonetheless, one can easily imagine that the level of emotions would very likely increase in the face of a new terrorist attack or earthquake (e.g., when hearing about a new negative event in the news or when even witnessing or experiencing such an event). Still, the expected difference between terrorist attacks and earthquakes should hold in this case as well.”

Similar for Study 2, plus the data presenting threat appraisal from lines 292-301 it is inconsistent with results from Study 1. For example “The terrorist attacks elicited more threat appraisal (M = 0.50, SD = 2.92) than the earthquakes (M = -0.39, SD = 2.76), which supported Hypothesis 1.” Means and standard deviations are not properly presented, a mean is not supposed to be negative under the presented methodology. Please revise this section.

We have also extended the discussion section of Study 2 to address the low to medium levels of anger, sadness, and anxiety.

“Similar to Study 1, the scores of the emotions anger, sadness, and anxiety were around the midpoint or lower end of the 9-point scale. As explained in the discussion of Study 1, these moderate scores may have occurred as participants had merely read about the negative events. Possibly, hearing about or watching news on television about new negative events, or even witnessing or experiencing these events may intensely increase these emotions. Still, the difference between earthquakes and terrorist attacks should also hold in such situations.”

In the case of the threat appraisal and its results, we apologize for not being clear enough about the measure. We measured threat appraisal with two items and then calculated a difference score between these items, which resulted in a single score for threat appraisal. Therefore, this score could take positive as well as negative values. In order to make up for this shortcoming, we have extended the description of the threat appraisal measure in the methods section of Study 2 and describe in more detail that and how we calculated the difference score.

“Threat appraisal was measured with two items, a demands item and a resources item. The demands item asked participants “How demanding is the negative event in the article you just read for you?” and assessed their answers on a 9-point Likert scale ranging from 1 (not demanding at all) to 9 (very demanding). The resources item asked participants “How well can you deal with the negative event?” and assessed their answers on a 9-point Likert scale ranging from 1 (not at all) to 9 (very well). These items are typical items often used in research about threat [27, 40]. We then calculated a difference score of the two items by subtracting the scores of the resources item from the scores of the demands item, which resulted in a single score for threat appraisal. Thus, this difference score could take positive as well as negative values with higher values indicating more threat appraisal.”

Overall, it is an important study, and should be considered for publication in PLOS ONE, once the statistical issue has been resolved.

Thank you very much again for your overall positive feedback. We hope that we have revised the paper in accordance with what you had in mind. Thank you also very much again for the time and effort you invested in reviewing our manuscript. The revisions have improved the manuscript and we think that it is ready for publication.

All the best and kind regards,

Hannah Greving

---

## [Decision Letter · Decision Letter 1]

31 Jan 2020

PONE-D-19-26386R1

The impact of event type and geographical proximity on threat appraisal and emotions in Wikipedia articles

PLOS ONE

Dear Dr. Greving,

Thank you for submitting your manuscript to PLOS ONE. After careful consideration, we feel that it has merit but does not fully meet PLOS ONE’s publication criteria as it currently stands. Therefore, we invite you to submit a revised version of the manuscript that addresses the points raised during the review process.

There is feedback from a Reviewer who provided extensive comments about this version of the manuscript.  Please, see the specific comments appended at the bottom of this letter. As you will see, there were several major concerns that should be addressed in another reviewed version of your study. Please, notice that because this can be considered as a major review, a resubmission will require another round of reviews and that the outcome is unpredictable at this point. 

We would appreciate receiving your revised manuscript by Mar 16 2020 11:59PM. To enhance the reproducibility of your results, we recommend that if applicable you deposit your laboratory protocols in protocols.io, where a protocol can be assigned its own identifier (DOI) such that it can be cited independently in the future. For instructions see: http://journals.plos.org/plosone/s/submission-guidelines#loc-laboratory-protocols

We look forward to receiving your revised manuscript.

Kind regards,

Angel Blanch, Ph.D.

Academic Editor

PLOS ONE

Reviewers' comments:

Reviewer's Responses to Questions

**Comments to the Author**

1. If the authors have adequately addressed your comments raised in a previous round of review and you feel that this manuscript is now acceptable for publication, you may indicate that here to bypass the “Comments to the Author” section, enter your conflict of interest statement in the “Confidential to Editor” section, and submit your "Accept" recommendation.

Reviewer #1: (No Response)

2. Is the manuscript technically sound, and do the data support the conclusions?

Reviewer #1: Yes

3. Has the statistical analysis been performed appropriately and rigorously? 

Reviewer #1: I Don't Know

4. Have the authors made all data underlying the findings in their manuscript fully available?

Reviewer #1: Yes

5. Is the manuscript presented in an intelligible fashion and written in standard English?

Reviewer #1: Yes

6. Review Comments to the Author

Reviewer #1: This short paper reports on two related studies of emotion expression/perception in Wikipedia articles on earthquakes and terrorist attacks, in a continuation of the authors’ recent research in this area. I agreed to review it because the abstract caught my interest, particular the geographical findings. All the findings are commonsensical and more-or-less predictable, but it can be gratifying to have one’s perceptions confirmed. At first view, the paper appears to be a straightforward and concisely-presented empirical study. It is focused on the research questions, methods, and results of the two related studies, with minimal conceptual framing, review of previous literature, or interpretation beyond confirming or disconfirming the studies’ hypotheses. Although this may be a genre that is acceptable for this journal, I generally see these lacks as shortcomings. In the case of this paper, the abbreviated literature review is most problematic, because readers are not told enough about the authors’ previous work to be able to judge what is original and new in the present work. The References indicate that the authors have published three recent articles (in 2018 and 2019) on what appear to be very closely-related topics. Therefore, the paper needs to make the unique contributions of the present study crystal clear. The geographical question seems to be new in this work; however, the originality of the other research questions is less clear.

I wrote “at first sight” above, because as I read through the paper, I became increasingly aware that two key points are in fact presented somewhat confusingly. The first could be a terminological issue; when I read “the current research coded the emotional content of existing Wikipedia articles” and that the authors “conducted a classical coding study,” I understood that they used content analysis to code emotional expression in the Wikipedia articles. I was surprised to learn later that what they meant was that they had study participants rate their emotional responses to the articles on Likert scales. In my understanding this doesn’t involve “coding,” but rather “rating.”

There is also a deeper underlying confusion that runs through the paper, between production, manifest content, and reception. The authors actually studied the participants’ emotional responses (reception), but they describe it as coding “the emotional content of existing Wikipedia articles” (manifest content). Moreover, later in the paper, they write that “threat appraisal and anger effects occurred … while externalizing information (i.e., contributing information to Wikipedia articles).” This would be fine if Wikipedia contributors had been studied, but they were not; the authors seem to assume that the emotions manifest in the articles are identical to the emotions of the producers of the articles. (The present paper doesn’t analyze the manifest content of the articles, but it seems as though one of their other papers may have done so. Either that, or the authors are assuming that the emotions expressed in the articles are the same as the raters’ emotional responses to the articles, in which case the only emotions for which there is empirical evidence are those of the raters.) In any case, the distinction between production, manifest content, and reception needs to be made clear. Both studies in this paper are concerned primarily with reception.

A third criticism is that the authors describe Study 1 and Study 2 as neatly complementary, “focusing either on many articles with a few coders or a few articles with many coders,” without discussing or even acknowledging in the discussion that the two studies were conducted in different languages, with different participants and different Wikipedia articles. These differences may not ultimately matter much to the studies’ findings, but their possible effects should be considered.

The paper is written in good, clear English overall. I have highlighted some awkward, nonnative wording in the attached copy of the manuscript.

More detailed comments follow:

L 38 Is www.alexa.com a credible scholarly source? It’s not in the references.

Ll 56-58 and other statements led me to believe that the content of the articles was coded (using content analysis). But later (e.g., ll 158-160) it seems that it is coders’ emotional reactions that are analyzed, which is not the same thing. Clarify.

Ll 113-114 “LIWC can be criticized for being based merely on the counts of words, without being able to assess broader measures of the complex categories it assesses.” (Vague. What broader measures? For example, one of LIWC’s main weaknesses is its insensitivity to the context in which the words appear, which can affect their meaning.)

Ll 127-129 Are you hypothesizing this, or stating it as a fact? If the latter, does that make your research results a foregone conclusion?

Ll 133-138 Were the four articles in Study 2 included in the 60 articles Study 1?

Ll 144-149 Don’t you already know the answers for some of these Hypotheses from your previous research? What is new about these here?

Ll 158-160 “The elicitation of emotions while reading the articles was measured by having four independent coders rate their subjective emotional reactions during reading.” (Mention this earlier. I misunderstood that the four coders were content analyzing the text of the articles.)

Ll 170-173 From among all entries that met these criteria for inclusion, how did you choose the 30 (60) for this study? Presumably there were not exactly 30 of each type that met your criteria.

Ll 190-201 Consistent with your previous research findings? Indicate which of these findings are previously unreported.

Ll 226-227 “Study 2 used four existing Wikipedia articles about an earthquake in Asia, an earthquake in Europe, a terrorist attack in Asia, and a terrorist attack in Europe.” (How were these articles selected?)

Ll 236-239 “This analysis showed that we would need a total sample size of 10 participants. We followed these calculations, but simultaneously had to adjust to the conditions in the laboratory and the availability of participants who signed up for the study. Therefore, 35 participants …” (How does it follow that you got 35 if you only needed 10? Explain better.)

Ll 241-242 Was the first study in the lab session related to this study?

Ll 250-252 “The criteria for the selection of the articles …” (Why did you switch from English-language articles in Study 1 to German articles in Study 2? It introduces another possible dimension of variation when comparing the two studies.)

Ll 254-256 How were these specific four articles selected from among all candidates that met your inclusion criteria?

Ll 350-352 “these emotions were elicited while reading as well as after reading these articles. …” (But not the same articles, in the same language, or by the same readers. Discuss.)

L 356 “The broader complexity of emotions” is vague. What do you mean by this, and how do your methods address emotional complexity? It seems that the same emotions are addressed as in previous research, and they are rated as conceptual wholes, without nuance as regards context, individual differences, etc. Expressing vs. perceiving emotions are complementary aspects of communication, to be sure, but do you mean to imply that emotions that are are both expressed and perceived are more “complex” than emotions are are expressed?

Ll 370-371 “threat appraisal and anger effects occurred both while externalizing information (i.e., contributing information to Wikipedia articles) and internalizing information (i.e., reading and …” (Careful of your assumptions. What Wikipedia authors write is not necessarily the same as what they feel. You cannot simply equate the two.)

Ll 384-385 “the anger effects for man-made attacks seemed to be independent from geographical proximity.” (Why might this be so? Is terrorism more “global” than earthquakes, perhaps, in an era where terrorists increasingly travel to wreak havok in other countries?)

Ll 388-389 “appraisal theories of specific emotions [24-26], theories of threat appraisal [27], and influences of emotions on information processing [16-17]” (These theories should be laid out more clearly early in the paper, rather than simply referenced.)

Ll 404-405 “The present research used and analyzed existing Wikipedia articles and coded them in two studies …” (Wording like this has caused me confusion throughout the article. You did not code the articles; you had readers rate their emotional responses to the articles, as I now understand. Clarify this throughout. The problem centers on the verb ‘code.’)

Ll 405-407 Your discussion and conclusions do not acknowledge the differences between the studies (different articles, different language of articles, different readers, different method of recording reactions, etc.). These should be discussed.

Ll 417-418 “the studies of this research were coding studies which used a classical coding approach” (I understand this as referring to content analysis, the classic coding and counting approach for analyzing textual and other content. What you did was conduct experiments and have participants rate their emotional responses to text.)

Ll 428-429 “These and similar measures would certainly be relevant measures in future research.” (Sentence is rather weak. What additional information might such studies provide? My guess is they will show the same patterns.)

---

7. PLOS authors have the option to publish the peer review history of their article (what does this mean?). If published, this will include your full peer review and any attached files.

Reviewer #1: No

---

## [Author Response · Author response to Decision Letter 1]

21 Feb 2020

Dear Dr. Angel Blanch,

Thank you very much for investing your time and effort to review our manuscript. Your constructive and valuable feedback helped us very much improve the paper on its way to publication.

We have considered all your suggestions and are convinced that we have made appropriate improvements. In this letter, we address all your comments and explain our revisions in detail. In order to ensure the transparency of our revisions, we have replied to each comment separately (in bold letter type; see attached Word document). Below our comments, you find the new or modified paragraphs of the revised manuscript (indented, in quotation marks, and in bold letter type; see attached Word document).

Reviewer #1: This short paper reports on two related studies of emotion expression/perception in Wikipedia articles on earthquakes and terrorist attacks, in a continuation of the authors’ recent research in this area. I agreed to review it because the abstract caught my interest, particular the geographical findings. All the findings are commonsensical and more-or-less predictable, but it can be gratifying to have one’s perceptions confirmed. At first view, the paper appears to be a straightforward and concisely-presented empirical study. It is focused on the research questions, methods, and results of the two related studies, with minimal conceptual framing, review of previous literature, or interpretation beyond confirming or disconfirming the studies’ hypotheses. Although this may be a genre that is acceptable for this journal, I generally see these lacks as shortcomings. In the case of this paper, the abbreviated literature review is most problematic, because readers are not told enough about the authors’ previous work to be able to judge what is original and new in the present work. The References indicate that the authors have published three recent articles (in 2018 and 2019) on what appear to be very closely-related topics. Therefore, the paper needs to make the unique contributions of the present study crystal clear. The geographical question seems to be new in this work; however, the originality of the other research questions is less clear.

We apologize for having presented only a minimal literature overview of the previous studies and related theories (see below for the other comment concerning the theories). To make up for this shortcoming, we have extended the introduction section of the manuscript and have presented more details about the previous research and the theories:

“Second, emotions are core to human beings and influence a wide array of aspects of human functioning and behavior [14]. In particular, they influence information processing [15-16], such as the production of information [17-18], which is relevant for writing Wikipedia articles. Across a variety of contexts, individuals produced more information that was congruent with their current state, for example, more negative information when they were in a negative state [18].” […]

“This research used appraisal theories of specific emotions [25-27] to explain the elicitation of certain emotions by certain negative events and used man-made attacks (e.g., terrorist attacks) and natural disasters (e.g., earthquakes) as negative events. These appraisal theories suggest that the characteristics of a situation are important for which emotions are elicited. Emotions can be elicited by desirable or undesirable outcomes, actions, or objects. As we investigate negative events, we will focus on undesirable outcomes and actions. Undesirable outcomes that cannot be made undone and have unspecific causes elicit sadness [25–27]. In contrast, undesirable outcomes that are caused by harmful and blameworthy actions of others who intended to do harm and can be held responsible elicit anger. The previous research mapped these predictions of appraisal theories onto the natural disasters and man-made attacks. In particular, terrorist attacks were expected to elicit more anger, as they occur due to the blameworthy actions of extremists, while earthquakes were expected to elicit more sadness, as they occur due to natural factors and cannot be undone [25-27] (for more elaboration on these expectations see [5]). Following this theorizing and these expectations, the previous research analyzed existing Wikipedia articles in order to investigate whether those expected emotional reactions were present in the manifest content of the articles. It was found that Wikipedia articles on man-made attacks contained more anger-related content than Wikipedia articles on natural disasters, whereas Wikipedia articles on natural disasters contained more sadness-related content than Wikipedia articles on man-made attacks [4-5].

Besides the research on the content of existing Wikipedia articles, there has also been research that investigated whether these effects hold when individuals spontaneously produce a Wikipedia article about these negative events. That research found that participants also produced more anger-related content in Wikipedia articles on man-made attacks than in articles on natural disasters in controlled laboratory experiments [3]. These laboratory experiments further examined what could explain the robust anger effects. Based on theories of threat [28], it was argued that man-made attacks elicit threat appraisal and as a consequence anger. Threat appraisal is defined as feelings of being unable to deal with a demanding situation [29]. Threat theories suggest that threat elicits active and engaged reactions, such as increased behavioral intentions and self-serving reactions [28]. It has also been shown that terrorist attacks were perceived as threatening and increased support for engagement in war activities and aggressive reactions [30–32]. Based on these theories and findings, the previous research has demonstrated that man-made attacks elicited more threat appraisal than nature-made disasters. Threat appraisal was also a mediator that could explain the effects on anger and anger-related content in Wikipedia articles.”

Moreover, we have added more information to the introduction in which we point out the contribution of the previous research and especially the contribution of the current research more explicitly:

“This previous research investigated which manifest content existing Wikipedia articles about negative events have and which information is produced when spontaneously writing those Wikipedia articles. Thus, the research focused on the expression of emotions in the articles. What has been neglected so far is how information from the Wikipedia articles is received and which emotional reactions are perceived when reading them. These steps are quite important elements in the communication process of Wikipedia, as Wikipedia articles are produced and written for other Internet users who receive the information and perceive its emotional content.” […]

“Although widely used, LIWC can be criticized for being based merely on the counts of words. LIWC is not able to assess any context that these emotional words are presented in, any relationships between emotional words, which may hint to totally different emotions (e.g., in the case of sarcasm or irony), or any political or societal meaning of acts or words (e.g., in the case of terrorist attacks). Thus, the results obtained in previous research for the emotional content in the Wikipedia articles may not completely hold, because LIWC may not entirely capture anger, sadness, and anxiety and may not consider the context or other relevant factors. It is also unclear whether individuals reading the Wikipedia articles recognize those emotions that LIWC found and react emotionally in the same way. In order to close this gap, the current research set out to investigate how the content and especially the emotional content of Wikipedia articles on negative events is perceived.”

I wrote “at first sight” above, because as I read through the paper, I became increasingly aware that two key points are in fact presented somewhat confusingly. The first could be a terminological issue; when I read “the current research coded the emotional content of existing Wikipedia articles” and that the authors “conducted a classical coding study,” I understood that they used content analysis to code emotional expression in the Wikipedia articles. I was surprised to learn later that what they meant was that they had study participants rate their emotional responses to the articles on Likert scales. In my understanding this doesn’t involve “coding,” but rather “rating.”

We regret that our wording has created a misunderstanding. To address this comment, we have removed the terms “code/coder/coding” from the entire manuscript and instead use “rate/rater/rating”, which are indeed better descriptions of what the raters/participants have done in the studies.

There is also a deeper underlying confusion that runs through the paper, between production, manifest content, and reception. The authors actually studied the participants’ emotional responses (reception), but they describe it as coding “the emotional content of existing Wikipedia articles” (manifest content). Moreover, later in the paper, they write that “threat appraisal and anger effects occurred … while externalizing information (i.e., contributing information to Wikipedia articles).” This would be fine if Wikipedia contributors had been studied, but they were not; the authors seem to assume that the emotions manifest in the articles are identical to the emotions of the producers of the articles. (The present paper doesn’t analyze the manifest content of the articles, but it seems as though one of their other papers may have done so. Either that, or the authors are assuming that the emotions expressed in the articles are the same as the raters’ emotional responses to the articles, in which case the only emotions for which there is empirical evidence are those of the raters.) In any case, the distinction between production, manifest content, and reception needs to be made clear. Both studies in this paper are concerned primarily with reception.

These are indeed valid points. First of all, we have given more details of our previous research in the introduction section (see above). This previous research had analyzed the manifest content of Wikipedia articles which is why we had drawn that conclusion. Second, we have added a paragraph to the introduction in which we disentangle and summarize where we have investigated manifest content and the production of information. We also clearly state that the current research deals with the reception of information and perception of emotional content (see above).

A third criticism is that the authors describe Study 1 and Study 2 as neatly complementary, “focusing either on many articles with a few coders or a few articles with many coders,” without discussing or even acknowledging in the discussion that the two studies were conducted in different languages, with different participants and different Wikipedia articles. These differences may not ultimately matter much to the studies’ findings, but their possible effects should be considered.

These are indeed relevant differences. We added a paragraph to the general discussion in which we discuss them:

“Still, some limitations can be discussed. First, the articles in Study 1 were in English to ensure comparability with previous research [4–5], while the articles of Study 2 were in German in order to keep the material of the entire laboratory study in German. Such language differences may have caused different results. The raters of the two studies were also different. While they were research assistants collaborating with the authors in Study 1, they were usual laboratory participants in Study 2. Yet, all raters of Study 1 and most raters of Study 2 were university students and therefore comparable. Moreover, the Wikipedia articles were different across the studies. There were more articles in Study 1 than in Study 2. Still, from the four articles of Study 2, one earthquake and one terrorist attack were also used in Study 1. In that sense, there was some overlap in terms of Wikipedia articles between the studies. Besides these differences, the two studies worked well together, but future research should nonetheless strive to minimize possible differences.”

The paper is written in good, clear English overall. I have highlighted some awkward, nonnative wording in the attached copy of the manuscript.

Thank you. We have changed these wordings accordingly.

More detailed comments follow:

L 38 Is www.alexa.com a credible scholarly source? It’s not in the references.

This website is a web tracking company and, thus, provides reliable statistics about Internet use. To give a more scientific source, we added a reference from the pew research center.

Ll 56-58 and other statements led me to believe that the content of the articles was coded (using content analysis). But later (e.g., ll 158-160) it seems that it is coders’ emotional reactions that are analyzed, which is not the same thing. Clarify.

We apologize for the misunderstanding. We have replaced the terms “code/coders/coding” in the entire manuscript with “rate/raters/rating” (see also above).

Ll 113-114 “LIWC can be criticized for being based merely on the counts of words, without being able to assess broader measures of the complex categories it assesses.” (Vague. What broader measures? For example, one of LIWC’s main weaknesses is its insensitivity to the context in which the words appear, which can affect their meaning.)

This is absolutely true. We have modified the paragraph about the limitations of LIWC and have added statements about the context problem and two more problematic aspects:

“Although widely used, LIWC can be criticized for being based merely on the counts of words. LIWC is not able to assess any context that these emotional words are presented in, any relationships between emotional words, which may hint to totally different emotions (e.g., in the case of sarcasm or irony), or any political or societal meaning of acts or words (e.g., in the case of terrorist attacks). Thus, the results obtained in previous research for the emotional content in the Wikipedia articles may not completely hold, because LIWC may not entirely capture anger, sadness, and anxiety and may not consider the context or other relevant factors. It is also unclear whether individuals reading the Wikipedia articles recognize those emotions that LIWC found and react emotionally in the same way. In order to close this gap, the current research set out to investigate how the content and especially the emotional content of Wikipedia articles on negative events is perceived."

Ll 127-129 Are you hypothesizing this, or stating it as a fact? If the latter, does that make your research results a foregone conclusion?

We did not intend to state this statement as a fact. To clarify, we have adjusted these sentences accordingly.

Ll 133-138 Were the four articles in Study 2 included in the 60 articles Study 1?

Two articles (April 2015 Nepal earthquake, January 2016 Bacha Khan University attack) were included in the 60 articles in Study 1. In the revised manuscript, we provide this information in the limitations section.

Ll 144-149 Don’t you already know the answers for some of these Hypotheses from your previous research? What is new about these here?

This is a valid point. Some of the hypotheses are not completely new. In the revised manuscript we state that directly prior to the hypotheses. What is new is the method with which we have tested the hypotheses. Therefore, before conducting the studies, it was not clear whether they would hold. And in case of sadness (Hypothesis 3), the hypothesis did not hold indeed.

Ll 158-160 “The elicitation of emotions while reading the articles was measured by having four independent coders rate their subjective emotional reactions during reading.” (Mention this earlier. I misunderstood that the four coders were content analyzing the text of the articles.)

We regret the misunderstanding. We have added this information in the abstract, the second paragraph of the introduction, and in “the current research” section.

Ll 170-173 From among all entries that met these criteria for inclusion, how did you choose the 30 (60) for this study? Presumably there were not exactly 30 of each type that met your criteria.

This is true. We first applied the selection criteria to the lists of earthquakes and terrorist attacks given in Wikipedia. Then, we applied the power calculations. And after that, we chose to use the 30 most recent events from the list in order to ensure timeliness and relevance of the events as well as being safe regarding the power of the study. We have also added this information to the text.

Ll 190-201 Consistent with your previous research findings? Indicate which of these findings are previously unreported.

We have added information about which result has been found in some but not other research.

Ll 226-227 “Study 2 used four existing Wikipedia articles about an earthquake in Asia, an earthquake in Europe, a terrorist attack in Asia, and a terrorist attack in Europe.” (How were these articles selected?)

We regret that we have not been clearer about how the articles were selected. We have provided more information about the additional selection criteria that we used:

“The criteria for the selection of the articles were the same as in Study 1, but we applied the following additional criteria: The articles were available in German, the lengths of the main texts were comparable across geographical location, and the event had happened in the last few years. These additional criteria yielded a sample of roughly 10 articles. From these articles, we then chose those four articles whose length of the German texts was sufficiently long to ensure that participants could engage with the negative events long enough.”

Ll 236-239 “This analysis showed that we would need a total sample size of 10 participants. We followed these calculations, but simultaneously had to adjust to the conditions in the laboratory and the availability of participants who signed up for the study. Therefore, 35 participants …” (How does it follow that you got 35 if you only needed 10? Explain better.)

We have explained in more detail for what reasons we had 35 participants.

“This analysis showed that we would need a total sample size of 10 participants. We followed these calculations, but simultaneously had to adjust to the conditions in the laboratory and the availability of participants. As participants had signed up for a whole lab session (see below), we could not simply stop the study after the 10 necessary participants. Therefore, more participants than needed participated in this study, that is, 35 participants (26 female, 9 male; Mage = 26.91, SD = 9.94) who had German as mother tongue took part in the study.”

Ll 241-242 Was the first study in the lab session related to this study?

The first and the second study were unrelated. The first study had no effects on the dependent measures of the second study.

Ll 250-252 “The criteria for the selection of the articles …” (Why did you switch from English-language articles in Study 1 to German articles in Study 2? It introduces another possible dimension of variation when comparing the two studies.)

We provide more elaboration on this point:

“We provided German articles as we wanted to keep the language of the laboratory study entirely in German. Moreover, the German articles may have been more easily comprehensible for participants.”

Moreover, we chose German as language as participants were to rate their emotional reactions after reading the text. By giving the texts in German, it was more likely that they remembered what they had read about and could more easily provide their emotional reactions.

Ll 254-256 How were these specific four articles selected from among all candidates that met your inclusion criteria?

We have provided more information about the additional selection criteria (see above).

Ll 350-352 “these emotions were elicited while reading as well as after reading these articles. …” (But not the same articles, in the same language, or by the same readers. Discuss.)

We have discussed these differences in the general discussion (see above).

L 356 “The broader complexity of emotions” is vague. What do you mean by this, and how do your methods address emotional complexity? It seems that the same emotions are addressed as in previous research, and they are rated as conceptual wholes, without nuance as regards context, individual differences, etc. Expressing vs. perceiving emotions are complementary aspects of communication, to be sure, but do you mean to imply that emotions that are are both expressed and perceived are more “complex” than emotions are are expressed?

We agree that this term was indeed confusing. We have removed it from the paper. In the introduction, we now describe more precisely what LIWC may not be able to measure and pick that up again in the beginning of the general discussion. In the introduction, we have added a paragraph in which we refer to the distinction between expressing and perceiving emotions and their relevance for communication processes in Wikipedia.

Ll 370-371 “threat appraisal and anger effects occurred both while externalizing information (i.e., contributing information to Wikipedia articles) and internalizing information (i.e., reading and …” (Careful of your assumptions. What Wikipedia authors write is not necessarily the same as what they feel. You cannot simply equate the two.)

This is true and we did not mean to equate them. What we meant was the combined effects of the current research and the previous research. We have added this information and explain it more explicitly.

Ll 384-385 “the anger effects for man-made attacks seemed to be independent from geographical proximity.” (Why might this be so? Is terrorism more “global” than earthquakes, perhaps, in an era where terrorists increasingly travel to wreak havok in other countries?)

We are grateful for this valuable idea. We have added it to the text:

“Possibly, the fact that extremists increasingly aim to do harm across the whole world makes terrorist attacks a more global phenomenon than earthquakes that are tied to specific regions. These differences may have caused the particular anger effects.”

Ll 388-389 “appraisal theories of specific emotions [24-26], theories of threat appraisal [27], and influences of emotions on information processing [16-17]” (These theories should be laid out more clearly early in the paper, rather than simply referenced.)

We agree. We describe these theories in more detail in the introduction now (see above).

Ll 404-405 “The present research used and analyzed existing Wikipedia articles and coded them in two studies …” (Wording like this has caused me confusion throughout the article. You did not code the articles; you had readers rate their emotional responses to the articles, as I now understand. Clarify this throughout. The problem centers on the verb ‘code.’)

We regret that we have caused this confusion. We refer to it as rating now throughout the entire manuscript (see also above).

Ll 405-407 Your discussion and conclusions do not acknowledge the differences between the studies (different articles, different language of articles, different readers, different method of recording reactions, etc.). These should be discussed.

We have provided a discussion of these differences in the general discussion (see also above).

Ll 417-418 “the studies of this research were coding studies which used a classical coding approach” (I understand this as referring to content analysis, the classic coding and counting approach for analyzing textual and other content. What you did was conduct experiments and have participants rate their emotional responses to text.)

We did not intend to create confusion about what we did. We have removed the terms “code/coder/coding” from the manuscript and refer to it as “rate/rater/rating” now (see also above).

Ll 428-429 “These and similar measures would certainly be relevant measures in future research.” (Sentence is rather weak. What additional information might such studies provide? My guess is they will show the same patterns.)

This is a valid point. We have deleted this sentence and added the following sentence:

“If such future research found similar results as we did, it could strengthen the current findings and, therefore, provide further relevant physiological insights about emotional reactions to Wikipedia articles about negative events.”

Thank you very much again for your overall positive feedback. We hope that we have revised the paper in accordance with what you had in mind. Thank you also very much again for the time and effort you invested in reviewing our manuscript. The revisions have improved the manuscript and we hope you agree with us that it is ready for publication.

All the best and kind regards,

Hannah Greving

---

## [Decision Letter · Decision Letter 2]

14 Apr 2020

PONE-D-19-26386R2

The impact of event type and geographical proximity on threat appraisal and emotions in Wikipedia articles

PLOS ONE

Dear Dr. Greving,

Thank you for submitting your manuscript to PLOS ONE. After careful consideration, we feel that it has merit but does not fully meet PLOS ONE’s publication criteria as it currently stands. Therefore, we invite you to submit a revised version of the manuscript that addresses the points raised during the review process.

The manuscript has been reviewed by the same reviewer who did the initial evaluation of your study. As you will see in the comments appended below, there were still several concerns that should be addressed in another version of the study.

We would appreciate receiving your revised manuscript by May 29 2020 11:59PM. To enhance the reproducibility of your results, we recommend that if applicable you deposit your laboratory protocols in protocols.io, where a protocol can be assigned its own identifier (DOI) such that it can be cited independently in the future. For instructions see: http://journals.plos.org/plosone/s/submission-guidelines#loc-laboratory-protocols

We look forward to receiving your revised manuscript.

Kind regards,

Angel Blanch, Ph.D.

Academic Editor

PLOS ONE

Reviewers' comments:

Reviewer's Responses to Questions

**Comments to the Author**

1. If the authors have adequately addressed your comments raised in a previous round of review and you feel that this manuscript is now acceptable for publication, you may indicate that here to bypass the “Comments to the Author” section, enter your conflict of interest statement in the “Confidential to Editor” section, and submit your "Accept" recommendation.

Reviewer #1: (No Response)

2. Is the manuscript technically sound, and do the data support the conclusions?

Reviewer #1: Yes

3. Has the statistical analysis been performed appropriately and rigorously? 

Reviewer #1: Yes

4. Have the authors made all data underlying the findings in their manuscript fully available?

Reviewer #1: Yes

5. Is the manuscript presented in an intelligible fashion and written in standard English?

Reviewer #1: Yes

6. Review Comments to the Author

Reviewer #1: Most of the points I raised have been cleared up through the authors’ revisions. However, one conceptual confusion persists. Are the raters in Study 1 rating the emotion expressed in the articles (let’s call this A), or rating their own emotional response (call this B)? This confusion is present right from the beginning in the Abstract and the Introduction. Compare, for example, the following two quotes:

Ll 53-55 – “and whether the emotional content in the Wikipedia articles also elicits similar emotional reactions in its readers while and after reading the articles” (B)

Vs.

Ll 58-61 – “Therefore, the current research rated the emotional content of existing Wikipedia articles — with either few raters rating many Wikipedia articles or many raters rating few Wikipedia articles ...” (A)

In other places, (A) and (B) are treated as equivalent in the same text, e.g.,

Ll 212-213 “In order to rate the emotionality of the articles (A), we used four independent raters. Their task was to read each article and rate their own emotional reactions (B) while reading”

(A) and (B) are different. Readers’ emotional reactions are not the same as their perceptions of what emotions are expressed in the text. In fact, after reading the rest of the article, it becomes clearer that only (B) was investigated. To study (A), the authors would have had to have the raters rate the emotion expressed in the text, without regard to their own emotional reactions, e.g., as a content analysis.

Yet it seems the authors also want to claim (A), as a check/confirmation on a previous study that used LIWC to quantify the emotions expressed in the articles themselves. See, e.g.,

Ll 421-423 “The present research makes up for [LIWC’s] shortcoming and complements such previous research by having the Wikipedia articles rated (A) by individuals who were capable of taking those additional factors into account”

(A) would have directly complemented the LIWC study. But that is not what was done in this study. Therefore, all wording that implies that the authors’ did (A) should be changed or omitted.

Other comments:

L 83 – “a variety of contexts, [studies have shown that] individuals produce[d � 0] more information that was congruent with their …”

L 87 – “were reflected”  “are reflected”

L 98 “cannot be made undone”  “cannot be undone”

Ll 129-130 “how information from the Wikipedia articles is received and which emotional reactions are perceived when reading them.” ( (A?) Or do you mean, which emotional reactions are produced? That would be (B))

Ll 147-148 “unclear whether individuals reading the Wikipedia articles recognize those emotions that LIWC found (A) and react emotionally (B) in the same way. (These are two separate things.)

Ll 153-155 “only English Wikipedia articles were analyzed. This does not allow for taking into account how close or distant the negative event had been for the Internet users who contributed to these articles” (Why not? Because English is used around the world? Explain.)

Ll 166-169; 192-194 Focus is squarely on (B) here. Fine; that is what you did.

Ll 212-213 “In order to rate the emotionality of the articles (A), we used four independent raters. Their task was to read each article and rate their own emotional reactions (B) while reading” (You did not do (A).)

Ll 306-311 (Mention here that two articles were also in Study 1 (in English)?)

Ll 421-423 “The present research makes up for [LIWC’s] shortcoming and complements such previous research by having the Wikipedia articles rated (A) by individuals who were capable of taking those additional factors into account” (This study does not directly complement the LIWC study; that would be study (A), having readers rate the emotion expressed in the text (without regard to their own emotions).)

Ll 429-432 “The previous laboratory research also demonstrated that merely thinking about a hypothetical man-made attack elicited more anger and a higher threat appraisal than thinking about a hypothetical man-made or nature-made disaster.”

(If I understand what you wrote about this study earlier, you did not measure their emotional response directly, but rather you had participants write Wikipedia articles in which you measured the expressions of emotion, and from which you inferred their emotions indirectly. This is perhaps a subtle distinction, but it illustrates how you conflate emotional reaction and emotional expression. In the same individual, it is relatively unproblematic to equate the two. But in this study it seems you are trying to equate the emotional reactions of your participants with the emotional expression in articles written by others (which you are assuming is the same as their emotional reactions). This is less straightforward.)

Ll 435- 438 “the threat appraisal and anger effects occurred both while externalizing information (i.e., contributing information to Wikipedia articles) [3–5] and internalizing information (i.e., reading and encoding information from Wikipedia articles)” (Yes, this is a more neutral and accurate way to put it.)

Ll 486-487 “Still, from the four articles of Study 2, one earthquake and one terrorist attack were also used in Study 1.” (Thanks for clarifying this. It should also be noted in describing your sampling procedure for study 2.)

===

7. PLOS authors have the option to publish the peer review history of their article (what does this mean?). If published, this will include your full peer review and any attached files.

Reviewer #1: Yes: Susan C. Herring

---

## [Author Response · Author response to Decision Letter 2]

21 Apr 2020

Dear Dr. Blanch, dear Dr. Herring,

Thank you very much for investing your time and effort to re-review our manuscript. Your constructive and valuable feedback again helped us very much improve the paper.

We have considered all your suggestions and explain our revisions in detail. In order to ensure the transparency of our revisions, we have replied to each comment separately (in bold letter type). Below our comments, you find the new or modified paragraphs of the revised manuscript (indented, in quotation marks, and in bold letter type).

Reviewer #1: Most of the points I raised have been cleared up through the authors’ revisions. However, one conceptual confusion persists. Are the raters in Study 1 rating the emotion expressed in the articles (let’s call this A), or rating their own emotional response (call this B)? This confusion is present right from the beginning in the Abstract and the Introduction. Compare, for example, the following two quotes:

Ll 53-55 – “and whether the emotional content in the Wikipedia articles also elicits similar emotional reactions in its readers while and after reading the articles” (B)

Vs.

Ll 58-61 – “Therefore, the current research rated the emotional content of existing Wikipedia articles — with either few raters rating many Wikipedia articles or many raters rating few Wikipedia articles ...” (A)

In other places, (A) and (B) are treated as equivalent in the same text, e.g.,

Ll 212-213 “In order to rate the emotionality of the articles (A), we used four independent raters. Their task was to read each article and rate their own emotional reactions (B) while reading”

(A) and (B) are different. Readers’ emotional reactions are not the same as their perceptions of what emotions are expressed in the text. In fact, after reading the rest of the article, it becomes clearer that only (B) was investigated. To study (A), the authors would have had to have the raters rate the emotion expressed in the text, without regard to their own emotional reactions, e.g., as a content analysis.

Yet it seems the authors also want to claim (A), as a check/confirmation on a previous study that used LIWC to quantify the emotions expressed in the articles themselves. See, e.g.,

Ll 421-423 “The present research makes up for [LIWC’s] shortcoming and complements such previous research by having the Wikipedia articles rated (A) by individuals who were capable of taking those additional factors into account”

(A) would have directly complemented the LIWC study. But that is not what was done in this study. Therefore, all wording that implies that the authors’ did (A) should be changed or omitted.

We are grateful for the constructive feedback on the conflation between content coding (A) and emotional reactions to content (B). We regret that we have created confusion regarding this point. In the revised manuscript, we have pointed out very clearly that this is an article about emotional reactions to content. We went carefully through the entire manuscript and revised those parts that could cause confusion. We have also revised the statements you quoted above (see also below).

Other comments:

L 83 – “a variety of contexts, [studies have shown that] individuals produce[d � 0] more information that was congruent with their …”

L 87 – “were reflected”  “are reflected”

L 98 “cannot be made undone”  “cannot be undone”

We have changed these sentences accordingly, thank you very much.

Ll 129-130 “how information from the Wikipedia articles is received and which emotional reactions are perceived when reading them.” ( (A?) Or do you mean, which emotional reactions are produced? That would be (B))

We meant indeed the emotional reactions that are elicited when reading the articles. We have rephrased this statement accordingly.

Ll 147-148 “unclear whether individuals reading the Wikipedia articles recognize those emotions that LIWC found (A) and react emotionally (B) in the same way. (These are two separate things.)

This is true. We have changed the sentence accordingly and only refer to the case that individuals reading Wikipedia articles react emotionally in the same way as indicated by LIWC.

Ll 153-155 “only English Wikipedia articles were analyzed. This does not allow for taking into account how close or distant the negative event had been for the Internet users who contributed to these articles” (Why not? Because English is used around the world? Explain.)

We regret that we have not been clear enough concerning this point. We have explained in more detail that anyone who can write in English can contribute to the English articles, meaning that the language of the articles does not indicate where the negative event had happened:

“In this previous research, only English Wikipedia articles were analyzed. This means that the articles could have been written anywhere by anyone who speaks English. Consequently, English language articles do not allow for taking into account how close or distant the negative event had been for the Internet users who contributed to these articles.”

Ll 166-169; 192-194 Focus is squarely on (B) here. Fine; that is what you did.

This is true. We have done our best to clarify this point throughout the entire manuscript.

Ll 212-213 “In order to rate the emotionality of the articles (A), we used four independent raters. Their task was to read each article and rate their own emotional reactions (B) while reading” (You did not do (A).)

We have rephrased the sentence and clearly state that the raters rated their emotional reactions to the articles.

Ll 306-311 (Mention here that two articles were also in Study 1 (in English)?)

We have provided this information at the end of this paragraph.

Ll 421-423 “The present research makes up for [LIWC’s] shortcoming and complements such previous research by having the Wikipedia articles rated (A) by individuals who were capable of taking those additional factors into account” (This study does not directly complement the LIWC study; that would be study (A), having readers rate the emotion expressed in the text (without regard to their own emotions).)

This is a valid point. We have removed the claim to directly complement the LIWC study.

Ll 429-432 “The previous laboratory research also demonstrated that merely thinking about a hypothetical man-made attack elicited more anger and a higher threat appraisal than thinking about a hypothetical man-made or nature-made disaster.”

(If I understand what you wrote about this study earlier, you did not measure their emotional response directly, but rather you had participants write Wikipedia articles in which you measured the expressions of emotion, and from which you inferred their emotions indirectly. This is perhaps a subtle distinction, but it illustrates how you conflate emotional reaction and emotional expression. In the same individual, it is relatively unproblematic to equate the two. But in this study it seems you are trying to equate the emotional reactions of your participants with the emotional expression in articles written by others (which you are assuming is the same as their emotional reactions). This is less straightforward.)

We apology for this misunderstanding and indeed also think that the distinction between emotional reaction and emotional expression is important. The laboratory studies also assessed participants’ emotional reactions and threat appraisal with self-reported measures in addition to the emotional content in the Wikipedia articles written by the participants. To make up for it, we have added this crucial information in the introduction as well as in the discussion:

“That research found that participants also produced more anger-related content in Wikipedia articles on man-made attacks than in articles on natural disasters in controlled laboratory experiments [3]. These laboratory experiments also assessed participants’ emotional reactions with self-report measures and further examined what could explain the robust anger effects, both as content in the Wikipedia articles and as self-reported reaction. Based on theories of threat, […]. Based on these theories and findings, the previous research has demonstrated that man-made attacks elicited more threat appraisal than nature-made disasters. Threat appraisal was also a mediator that could explain the effects on anger and anger-related content in Wikipedia articles.”

“The previous laboratory research also measured emotional reactions and threat separately with self-report measures and demonstrated that merely thinking about a hypothetical man-made attack elicited more anger and a higher threat appraisal than thinking about a hypothetical man-made or nature-made disaster.”

Ll 435- 438 “the threat appraisal and anger effects occurred both while externalizing information (i.e., contributing information to Wikipedia articles) [3–5] and internalizing information (i.e., reading and encoding information from Wikipedia articles)” (Yes, this is a more neutral and accurate way to put it.)

Thank you.

Ll 486-487 “Still, from the four articles of Study 2, one earthquake and one terrorist attack were also used in Study 1.” (Thanks for clarifying this. It should also be noted in describing your sampling procedure for study 2.)

We have added this information to the methods section of Study 2, where we described the sampling procedure.

Thank you very much again for your constructive feedback. We hope that we have revised the paper in accordance with what you had in mind. Thank you also very much again for your time and effort. The revisions have improved the manuscript and we hope you agree with us that it is ready for publication.

All the best and kind regards,

Hannah Greving

---

## [Editor Report · Decision Letter 3]

5 May 2020

PONE-D-19-26386R3

The impact of event type and geographical proximity on threat appraisal and emotional reactions to Wikipedia articles

PLOS ONE

Dear Dr. Greving,

Thank you for submitting your manuscript to PLOS ONE. After careful consideration, we feel that it has merit but does not fully meet PLOS ONE’s publication criteria as it currently stands. Therefore, we invite you to submit a revised version of the manuscript that addresses the points raised during the review process.

The manuscript has been evaluated by the same reviewer who conducted the previous evaluation of your work. As you will see in the comments appended below, a concern persisted in this version of the manuscript regarding the evaluation of the emotions.

We would appreciate receiving your revised manuscript by Jun 19 2020 11:59PM. To enhance the reproducibility of your results, we recommend that if applicable you deposit your laboratory protocols in protocols.io, where a protocol can be assigned its own identifier (DOI) such that it can be cited independently in the future. For instructions see: http://journals.plos.org/plosone/s/submission-guidelines#loc-laboratory-protocols

We look forward to receiving your revised manuscript.

Kind regards,

Angel Blanch, Ph.D.

Academic Editor

PLOS ONE

---

## [Author Response · Author response to Decision Letter 3]

11 May 2020

Dear Dr. Blanch, dear Dr. Herring,

Thank you very much for investing your time and effort to re-review our manuscript. Your constructive and valuable feedback again helped us very much improve the paper.

We have considered all your suggestions and explain our revisions in detail. In order to ensure the transparency of our revisions, we have replied to each comment separately (in bold letter type). Below our comments, you find the new or modified paragraphs of the revised manuscript (indented, in quotation marks, and in bold letter type).

Reviewer #1: Most of the points I raised have been cleared up through the authors’ revisions. However, one conceptual confusion persists. Are the raters in Study 1 rating the emotion expressed in the articles (let’s call this A), or rating their own emotional response (call this B)? This confusion is present right from the beginning in the Abstract and the Introduction. Compare, for example, the following two quotes:

Ll 53-55 – “and whether the emotional content in the Wikipedia articles also elicits similar emotional reactions in its readers while and after reading the articles” (B)

Vs.

Ll 58-61 – “Therefore, the current research rated the emotional content of existing Wikipedia articles — with either few raters rating many Wikipedia articles or many raters rating few Wikipedia articles ...” (A)

In other places, (A) and (B) are treated as equivalent in the same text, e.g.,

Ll 212-213 “In order to rate the emotionality of the articles (A), we used four independent raters. Their task was to read each article and rate their own emotional reactions (B) while reading”

(A) and (B) are different. Readers’ emotional reactions are not the same as their perceptions of what emotions are expressed in the text. In fact, after reading the rest of the article, it becomes clearer that only (B) was investigated. To study (A), the authors would have had to have the raters rate the emotion expressed in the text, without regard to their own emotional reactions, e.g., as a content analysis.

Yet it seems the authors also want to claim (A), as a check/confirmation on a previous study that used LIWC to quantify the emotions expressed in the articles themselves. See, e.g.,

Ll 421-423 “The present research makes up for [LIWC’s] shortcoming and complements such previous research by having the Wikipedia articles rated (A) by individuals who were capable of taking those additional factors into account”

(A) would have directly complemented the LIWC study. But that is not what was done in this study. Therefore, all wording that implies that the authors’ did (A) should be changed or omitted.

We are grateful for the constructive feedback on the conflation between content coding (A) and emotional reactions to content (B). We regret that we have created confusion regarding this point. In the revised manuscript, we have pointed out very clearly that this is an article about emotional reactions to content. We went carefully through the entire manuscript and revised those parts that could cause confusion. We have also revised the statements you quoted above (see also below).

Other comments:

L 83 – “a variety of contexts, [studies have shown that] individuals produce[d � 0] more information that was congruent with their …”

L 87 – “were reflected”  “are reflected”

L 98 “cannot be made undone”  “cannot be undone”

We have changed these sentences accordingly, thank you very much.

Ll 129-130 “how information from the Wikipedia articles is received and which emotional reactions are perceived when reading them.” ( (A?) Or do you mean, which emotional reactions are produced? That would be (B))

We meant indeed the emotional reactions that are elicited when reading the articles. We have rephrased this statement accordingly.

Ll 147-148 “unclear whether individuals reading the Wikipedia articles recognize those emotions that LIWC found (A) and react emotionally (B) in the same way. (These are two separate things.)

This is true. We have changed the sentence accordingly and only refer to the case that individuals reading Wikipedia articles react emotionally in the same way as indicated by LIWC.

Ll 153-155 “only English Wikipedia articles were analyzed. This does not allow for taking into account how close or distant the negative event had been for the Internet users who contributed to these articles” (Why not? Because English is used around the world? Explain.)

We regret that we have not been clear enough concerning this point. We have explained in more detail that anyone who can write in English can contribute to the English articles, meaning that the language of the articles does not indicate where the negative event had happened:

“In this previous research, only English Wikipedia articles were analyzed. This means that the articles could have been written anywhere by anyone who speaks English. Consequently, English language articles do not allow for taking into account how close or distant the negative event had been for the Internet users who contributed to these articles.”

Ll 166-169; 192-194 Focus is squarely on (B) here. Fine; that is what you did.

This is true. We have done our best to clarify this point throughout the entire manuscript.

Ll 212-213 “In order to rate the emotionality of the articles (A), we used four independent raters. Their task was to read each article and rate their own emotional reactions (B) while reading” (You did not do (A).)

We have rephrased the sentence and clearly state that the raters rated their emotional reactions to the articles.

Ll 306-311 (Mention here that two articles were also in Study 1 (in English)?)

We have provided this information at the end of this paragraph.

Ll 421-423 “The present research makes up for [LIWC’s] shortcoming and complements such previous research by having the Wikipedia articles rated (A) by individuals who were capable of taking those additional factors into account” (This study does not directly complement the LIWC study; that would be study (A), having readers rate the emotion expressed in the text (without regard to their own emotions).)

This is a valid point. We have removed the claim to directly complement the LIWC study.

Ll 429-432 “The previous laboratory research also demonstrated that merely thinking about a hypothetical man-made attack elicited more anger and a higher threat appraisal than thinking about a hypothetical man-made or nature-made disaster.”

(If I understand what you wrote about this study earlier, you did not measure their emotional response directly, but rather you had participants write Wikipedia articles in which you measured the expressions of emotion, and from which you inferred their emotions indirectly. This is perhaps a subtle distinction, but it illustrates how you conflate emotional reaction and emotional expression. In the same individual, it is relatively unproblematic to equate the two. But in this study it seems you are trying to equate the emotional reactions of your participants with the emotional expression in articles written by others (which you are assuming is the same as their emotional reactions). This is less straightforward.)

We apologize for this misunderstanding and indeed also think that the distinction between emotional reaction and emotional expression is important. The laboratory studies also assessed participants’ emotional reactions and threat appraisal with self-reported measures in addition to the emotional content in the Wikipedia articles written by the participants. To make up for it, we have added this crucial information in the introduction as well as in the discussion:

“That research found that participants also produced more anger-related content in Wikipedia articles on man-made attacks than in articles on natural disasters in controlled laboratory experiments [3]. These laboratory experiments also assessed participants’ emotional reactions with self-report measures and further examined what could explain the robust anger effects, both as content in the Wikipedia articles and as self-reported reaction. Based on theories of threat, […]. Based on these theories and findings, the previous research has demonstrated that man-made attacks elicited more threat appraisal than nature-made disasters. Threat appraisal was also a mediator that could explain the effects on anger and anger-related content in Wikipedia articles.”

“The previous laboratory research also measured emotional reactions and threat separately with self-report measures and demonstrated that merely thinking about a hypothetical man-made attack elicited more anger and a higher threat appraisal than thinking about a hypothetical man-made or nature-made disaster.”

Ll 435- 438 “the threat appraisal and anger effects occurred both while externalizing information (i.e., contributing information to Wikipedia articles) [3–5] and internalizing information (i.e., reading and encoding information from Wikipedia articles)” (Yes, this is a more neutral and accurate way to put it.)

Thank you.

Ll 486-487 “Still, from the four articles of Study 2, one earthquake and one terrorist attack were also used in Study 1.” (Thanks for clarifying this. It should also be noted in describing your sampling procedure for study 2.)

We have added this information to the methods section of Study 2, where we described the sampling procedure.

Thank you very much again for your constructive feedback. We hope that we have revised the paper in accordance with what you had in mind. Thank you also very much again for your time and effort. The revisions have improved the manuscript and we hope you agree with us that it is ready for publication.

All the best and kind regards,

Hannah Greving

---

## [Editor Report · Decision Letter 4]

13 May 2020

The impact of event type and geographical proximity on threat appraisal and emotional reactions to Wikipedia articles

PONE-D-19-26386R4

Dear Dr. Greving,

We are pleased to inform you that your manuscript has been judged scientifically suitable for publication and will be formally accepted for publication once it complies with all outstanding technical requirements.

With kind regards,

Angel Blanch, Ph.D.

Academic Editor

PLOS ONE
---

## [Editor Report · Acceptance letter]

19 May 2020

PONE-D-19-26386R4 

The impact of event type and geographical proximity on threat appraisal and emotional reactions to Wikipedia articles 

Dear Dr. Greving:

I am pleased to inform you that your manuscript has been deemed suitable for publication in PLOS ONE. Congratulations! Your manuscript is now with our production department. 

With kind regards,

on behalf of

Dr. Angel Blanch 

Academic Editor

PLOS ONE